# Optimizing Detection of Kidney Transplant Injury by Assessment of Donor-Derived Cell-Free DNA via Massively Multiplex PCR

**DOI:** 10.3390/jcm8010019

**Published:** 2018-12-23

**Authors:** Tara K. Sigdel, Felipe Acosta Archila, Tudor Constantin, Sarah A. Prins, Juliane Liberto, Izabella Damm, Parhom Towfighi, Samantha Navarro, Eser Kirkizlar, Zachary P. Demko, Allison Ryan, Styrmir Sigurjonsson, Reuben D. Sarwal, Szu-Chuan Hseish, Chitranon Chan-On, Bernhard Zimmermann, Paul R. Billings, Solomon Moshkevich, Minnie M. Sarwal

**Affiliations:** 1University of California San Francisco, Department of Surgery, San Francisco, CA 94143, USA; tara.sigdel@ucsf.edu (T.K.S.); juliane.liberto@ucsf.edu (J.L.); izabella.damm@ucsfmedctr.org (I.D.); parhom.towfighi@ucsf.edu (P.T.); reuben.sarwal@ucsf.edu (R.D.S.); suehsieh2011@gmail.com (S.-C.H.); chitranon.chan-on@ucsf.edu (C.C.-O.); 2Natera, Inc., San Carlos, CA 94070, USA; facosta@natera.com (F.A.A.); tudorpconstantin@gmail.com (T.C.); sprins@natera.com (S.A.P.); snavarro@natera.com (S.N.); kirkizlar@gmail.com (E.K.); zdemko@natera.com (Z.P.D.); aryan@natera.com (A.R.); ssigurjonsson@natera.com (S.S.); bzimmermann@natera.com (B.Z.); pbillings@natera.com (P.R.B.); smoshkevich@natera.com (S.M.)

**Keywords:** cfDNA, kidney transplantation, rejection

## Abstract

Standard noninvasive methods for detecting renal allograft rejection and injury have poor sensitivity and specificity. Plasma donor-derived cell-free DNA (dd-cfDNA) has been reported to accurately detect allograft rejection and injury in transplant recipients and shown to discriminate rejection from stable organ function in kidney transplant recipients. This study used a novel single nucleotide polymorphism (SNP)-based massively multiplexed PCR (mmPCR) methodology to measure dd-cfDNA in various types of renal transplant recipients for the detection of allograft rejection/injury without prior knowledge of donor genotypes. A total of 300 plasma samples (217 biopsy-matched: 38 with active rejection (AR), 72 borderline rejection (BL), 82 with stable allografts (STA), and 25 with other injury (OI)) were collected from 193 unique renal transplant patients; dd- cfDNA was processed by mmPCR targeting 13,392 SNPs. Median dd-cfDNA was significantly higher in samples with biopsy-proven AR (2.3%) versus BL (0.6%), OI (0.7%), and STA (0.4%) (*p* < 0.0001 all comparisons). The SNP-based dd-cfDNA assay discriminated active from non-rejection status with an area under the curve (AUC) of 0.87, 88.7% sensitivity (95% CI, 77.7–99.8%) and 72.6% specificity (95% CI, 65.4–79.8%) at a prespecified cutoff (>1% dd-cfDNA). Of 13 patients with AR findings at a routine protocol biopsy six-months post transplantation, 12 (92%) were detected positive by dd-cfDNA. This SNP-based dd-cfDNA assay detected allograft rejection with superior performance compared with the current standard of care. These data support the feasibility of using this assay to detect disease prior to renal failure and optimize patient management in the case of allograft injury.

## 1. Introduction

Precision medicine and personalized tailoring of immunosuppressive drug regimens can improve the current state of organ transplant management [1]. Transplantation injuries may be delayed in detection, and therefore treated ineffectively, because diagnosis can be difficult and biopsy, an invasive and potentially morbid procedure, may be inconclusive. Though advances in immunosuppressive drugs, organ procurement methods, and human leukocyte antigen-typing have lowered the number of clinical- and biopsy-confirmed rejection episodes, sub-clinical rejection of kidney grafts remains a significant risk [2,3]. Kidney transplant management is particularly challenging owing to the lack of sensitivity and specificity of the serum creatinine assay, which, in addition to the late detection of transplant injuries, makes immunosuppression dosage and adjustment far from personalized [4,5]. Therefore, rapid and non-invasive detection and prediction of allograft injury/rejection holds promise for improving the post-transplantation management of patients who have received kidney allografts.

Diagnosis of renal transplant rejection is generally dependent on an increase in serum creatinine levels or its algorithmic derivative, estimated glomerular filtration rate (eGFR), which indicates altered renal filtration functioning. Methods of estimating kidney rejection in allograft recipients based on serum creatinine or eGFR, however, lack sufficient accuracy. Since there are many causes of the baseline drift in altered renal filtering in these patients, biopsy is required for definitive diagnosis. However, biopsies are invasive and costly procedures, which limit their use in clinical practice. Furthermore, biopsy results are often plagued by expert reader variance and can lead to delayed diagnosis of active rejection, after which irreversible organ damage may have occurred [6,7]. There is a current unmet need for a rapid, accurate, and noninvasive approach to detecting allograft rejection and/or injury—one which may require integration of the current “gold” standard morphological assessments with modern molecular diagnostic tools [8].

Donor-derived cell-free DNA (dd-cfDNA) detected in the blood of transplant recipients has been reported as a noninvasive marker to diagnose allograft injury/rejection [9,10,11,12], and holds promise for producing faster and more quantitative results compared with current diagnostic options. Recently, it was demonstrated that plasma dd-cfDNA fraction, typically between 0.3% and 1.2% in stable patients [13], can discriminate active rejection status from stable organ function in kidney transplant recipients [14]. Previously we validated the clinical application of a targeted, single nucleotide polymorphism (SNP)-based cell-free assay targeting greater than 10,000 loci as a successful screening tool for the detection of fetal chromosomal abnormalities [15,16,17] and show here that a similar approach targeting 13,392 SNPs can be used to evaluate differences in donor cfDNA burden in different transplant rejection injuries over time. This study uses a novel SNP-based mmPCR-next generation sequencing (NGS) methodology to measure dd-cfDNA in renal transplant recipients for the detection of allograft rejection/injury without prior knowledge of donor genotypes.

## 2. Materials and Methods

### 2.1. Study Design 

This was a retrospective analysis of blood samples from kidney transplant recipients who had transplant surgeries at the University of California at San Francisco (USCF) Medical Center. The study was approved by the institutional review board at the UCSF Medical Center. All patients provided written informed consent to participate in the research, in full adherence to the Declaration of Helsinki. The clinical and research activities being reported are consistent with the Principles of the Declaration of Istanbul as outlined in the Declaration of Istanbul on Organ Trafficking and Transplant Tourism.

### 2.2. Study Population and Samples

Male and female adult or young-adult patients received a kidney from related or unrelated living donors, or unrelated deceased donors. Plasma samples were obtained from an existing biorepository; time points of patient blood draw following transplantation surgery were either at the time of an allograft biopsy or at various pre-specified time intervals based on lab protocols. Typically, samples were biopsy-matched and had blood drawn at the time of clinical dysfunction and biopsy or at the time of protocol biopsy (at which time most patients did not have clinical dysfunction). In addition, some patients had serial post transplantation blood drawn as part of routine Internal Review Board approved bio-sampling studies. The selection of study samples was based on (a) adequate plasma being available, and (b) if the sample was associated with biopsy information. Among the full 300 sample cohort, 72.3% were drawn on the day of biopsy. Patients without biopsy-matched samples were excluded from the primary analyses. 

### 2.3. Biopsy Samples

All kidney biopsies were analyzed in a blinded manner by a UCSF pathologist and were graded by the 2017 Banff classification [18] for active rejection (AR); intragraft C4d stains were performed [19] to assess for acute humoral rejection [20]. Biopsies were not done in cases of active urinary tract infection (UTI) or other infections. Transplant “injury” was defined as a >20% increase in serum creatinine from its previous steady-state baseline value and an associated biopsy that was classified as either active rejection (AR), borderline rejection (BL), or other injury (OI) (e.g., drug toxicity, viral infection). Active rejection was defined, at minimum, by the following criteria: (1) T-cell-mediated rejection (TCMR) consisting of either a tubulitis (t) score >2 accompanied by an interstitial inflammation (i) score >2 or vascular changes (v) score >0; (2) C4d positive antibody-mediated rejection (ABMR) consisting of positive donor specific antibodies (DSA) with a glomerulitis (g) score >0/or peritubular capillaritis score (ptc) >0 or v > 0 with unexplained acute tubular necrosis/thrombotic micro angiopathy (ATN/TMA) with C4d = 2; or (3) C4d negative ABMR consisting of positive DSA with unexplained ATN/TMA with g + ptc ≥2 and C4d is either 0 or 1. Borderline change (BL) was defined by t1 + i0, or t1 + i1, or t2 + i0 without explained cause (e.g., polyomavirus-associated nephropathy (PVAN)/infectious cause/ATN). Other criteria used for BL changes were g > 0 and/or ptc > 0, or v > 0 without DSA, or C4d or positive DSA, or positive C4d without nonzero g or ptc scores. Normal (STA) allografts were defined by an absence of significant injury pathology as defined by Banff schema. 

### 2.4. dd-cfDNA Measurement in Blood Samples

Cell-free DNA was extracted from plasma samples using the QIAamp Circulating Nucleic Acid Kit (Qiagen) and quantified on the LabChip NGS 5k kit (Perkin Elmer, Waltham, MA, USA) following manufacturer’s instructions. Cell-free DNA was input into library preparation using the Natera Library Prep kit as previously described [21], with a modification of 18 cycles of library amplification to plateau the libraries. Purified libraries were quantified using LabChip NGS 5k as previously described [21]. Target enrichment was accomplished using massively multiplexed-PCR (mmPCR) using a modified version of a previously described method [22], with 13,392 single nucleotide polymorphisms (SNPs) targeted. Amplicons were then sequenced on an Illumina HiSeq 2500 Rapid Run, 50 cycles single end, with 10–11 million reads per sample. 

### 2.5. Statistical Analyses of dd-cfDNA and eGFR

In each sample, dd-cfDNA was measured and correlated with rejection status, and results were compared with eGFR. Where applicable, all statistical tests were two sided. Significance was set at *p* < 0.05. Because the distribution of dd-cfDNA in patients was severely skewed among the groups, data were analyzed using a Kruskal–Wallis rank sum test followed by Dunn multiple comparison tests with Holm correction [23,24]. eGFR (serum creatinine in mg/dL) was calculated as described previously for adult [25] and pediatric patients [26]. Briefly, eGFR = 186 × Serum Creatinine^−1.154^ × Age^−0.203^ × (1.210 if Black) × (0.742 if Female).

To evaluate the performance of dd-cfDNA and eGFR (mL/min/1.73m^2^) as rejection markers, samples were separated into an AR group and a non-rejection group (BL + STA + OI). Using this categorization, the following predetermined cut-offs were used to classify a sample as AR: >1% for dd-cfDNA [14] and <60.0 for eGFR [27].

To calculate the performance parameters of each marker (sensitivity, specificity, positive predictive value (PPV), negative predictive value (NPV), and area under the curve (AUC)), a bootstrap method was used to account for repeated measurements within a patient [28]. Briefly, at each bootstrap step, a single sample was selected from each patient; by assuming independence among patients, the performance parameters and their standard errors were calculated. This was repeated 10,000 times; final confidence intervals were calculated using the bootstrap mean for the parameter with the average of the bootstrap standard errors with standard normal quantiles. Standard errors for sensitivity and specificity were calculated assuming a binomial distribution; for PPV and NPV a normal approximation was used; and for AUC the DeLong method was used. Performance was calculated for all samples with a matched biopsy, including the sub-cohort consisting of samples drawn at the same time as a protocol biopsy. 

Differences in dd-cfDNA levels by donor type (living related, living non-related, and deceased non-related) were also evaluated. Significance was determined using the Kruskal–Wallis rank sum test as described above. Inter- and intra-variability in dd-cfDNA over time was evaluated using a mixed effects model with a logarithmic transformation on dd-cfDNA [29]; 95% confidence intervals (CI) for the intra- and inter-patient standard deviations were calculated using a likelihood profile method.

Post hoc analyses evaluated (a) different dd-cfDNA thresholds to maximize NPV (Appendix A) and (b) combined dd-cfDNA and eGFR to define an empirical rejection zone that may improve the PPV for AR diagnosis (Appendix A). 

All analyses were done using R 3.3.2 using the FSA (for Dunn tests), lme4 (for mixed effect modeling) and pROC (for AUC calculations) packages.

## 3. Results

### 3.1. Patients and Blood Samples

A total of 300 plasma samples were collected from 193 unique renal transplant recipients. Of these, 23 samples from 15 patients did not meet inclusion criteria and were excluded from analyses; this included samples collected within three days from transplant (15), and samples unable to be sequenced (8). Of the remaining 277 samples, 217 were biopsy-matched, including 38 collected from patients with biopsy-proven active rejection (AR), 72 with biopsy-proven borderline rejection (BL), 82 normal, stable allografts (STA), and 25 with a biopsy that indicated other injury (OI) (Figure 1). Of the 178 unique patients included in the study, 20% (35) were under 18 years of age; 30% (54) were between 18 and 40 years, and 50% (89) were older than 40 years of age at the time of first blood sample. 

Published data have shown that dd-cfDNA fractions in patients with AR are significantly higher than patients with non-rejection; however, these data have shown an inability of dd-cfDNA to detect all types of AR, specifically failing to detect TCMR [14]. In this data set, the performance of the assay to detect rejection was evaluated for all types of rejection combined (ABMR, TCMR), based on the assumption that elevated dd-cfDNA levels are indicative of ongoing damage to the transplanted organ, irrespective of the underlying biology of rejection. Therefore, the ability of the assay to detect AR versus non-rejection was calculated, where non-rejection was defined as all specimens that were classified as STA, BL, or OI. Additionally, the performance of the assay to discriminate AR from complete absence of injury (STA) was also evaluated. A summary of demographic information and sample characteristics are provided in Table 1. All pathology samples were read at UCSF by a single renal pathologist and rated according to the recently updated Banff criteria [18].

### 3.2. dd-cfDNA and eGFR in Kidney Transplant Recipients

The amount of dd-cfDNA was significantly higher in the circulating plasma of the AR group (median = 2.32%) compared with the non-rejection group (median = 0.47%, *p* < 0.0001) (Table 2, Appendix A). Additionally, the median level of dd-cfDNA was significantly higher in the AR group compared with all three individual non-rejection subgroups: BL group (0.58%), STA group (0.40%), and OI (0.67%, all comparisons, adj. *p* < 0.0001) (Figure 2A, Appendix A). That the dd-cfDNA burden was higher in the AR group as compared to the BL group indicates that dd-cfDNA fraction may be used to track the evolution of early injury to more established rejection, as well as any subsequent recovery. The differences between the levels of dd-cfDNA between any of the non-rejection subgroups (STA, BL, and OI) were not significant (Figure 2A; Appendix A).

In contrast to dd-cfDNA, eGFR scores did not have as much discriminatory ability for differentiating AR and individual non-rejection groups (Table 2, Appendix A). Overall, the median eGFR score in the AR group (45.67) was significantly lower than that observed in the non-rejection group (76.6, *p* < 0.0001) (Table 2 and Appendix A, Appendix A) and even lower compared to the STA group alone (104.5, adj. *p* < 0.0001) (Table 2 and Appendix A, Figure 2B). However, unlike the dd-cfDNA results, there was no difference in median eGFR scores between the AR and BL groups (45.67 vs. 55.99, adj. *p* = 0.461) (Table 2 and Appendix A; Figure 2B). Additionally, compared with the STA group, eGFR levels were significantly higher in the BL (55.99, adj. *p* < 0.0001) and OI (57.4, adj. *p* < 0.0001) groups (Table 2 and Appendix A, Figure 2B). 

### 3.3. Performance Estimates for Discriminatory Ability of Tests

With a dd-cfDNA cutoff of >1%, the mmPCR-NGS method had an 88.7% sensitivity (95% CI, 77.7–99.8%) and 72.6% specificity (95% CI, 65.4–79.8%) for detection of AR. Sensitivity and specificity values are shown over the range of dd-cfDNA cutoffs in Figure 3A. The AUC was 0.87 (95% CI, 0.80–0.95). Based on a 25% prevalence of rejection in an at-risk population, the positive predictive value (PPV) was projected to be 52.0% (95% CI, 44.7–59.2%) and the negative predictive value (NPV) was projected to be 95.1% (95% CI, 90.5–99.7%). 

Sensitivity and specificity were lower using eGFR (Figure 3B). Using an eGFR cutoff score <60 for AR, sensitivity and specificity values were 67.8% (95% CI, 51.3–84.2%) and 65.3% (57.6–73.0%), respectively, with an AUC of 0.74 (0.66–0.83). The projected PPV and NPV values of eGFR were 39.4% (31.6–47.3%) and 85.9% (75.9–92.2%), respectively. 

As a post hoc analysis, we also evaluated a combination of eGFR with dd-cfDNA. Although we do not have a large number of samples to train a combined model, we can still see potential empirical rejection zones. Samples with a very high eGFR score, for example, tend to correspond to non-rejection samples (Appendix A). Defining the active rejection zone to be dd-cfDNA level >1% and eGFR <100, and non-rejection to be dd-cfDNA level <1% or eGFR >100, the combined dd-cfDNA and eGFR markers correctly classified 32/38 (84.2%) AR samples, and 145/179 (81.0%) non-rejection samples. Meanwhile at an equivalent specificity of 81.0%, (using a cut off of 1.3% dd-cfDNA) the sensitivity of the dd-cfDNA marker alone was 82.3%. Therefore the combined biomarker approach appeared to add little or no value over cfDNA alone.

### 3.4. dd-cfDNA Performance in Unique Biopsy-Confirmed Subgroups

Among the biopsy-matched samples, 103 (47.5%) were biopsied for clinical reasons, whereas 114 (52.5%) were biopsied according to protocol (Table 3 and Appendix A). Figure 4 depicts sample dd-cfDNA levels among all subgroups; 85 (39.2%) had dd-cfDNA levels >1%. Of those, 22 (25.9%) were STA; the remainder were AR (33 (38.8%)), OI (10 (11.8%)), or BL (20 (23.5%)). Of the individual groups, 33 (86.8%) of the total AR samples and 22 (26.8%) of the total STA samples had dd-cfDNA levels above 1%. In comparison, 20 (27.8%) of the total BL samples and 10 (40.0%) of the total OI samples had dd-cfDNA levels above 1%. 

Figure 4 shows assay performance for the subset of samples drawn at the time of a for-cause biopsy (4A) and protocol biopsy (4B); performance shown in protocol biopsies is expected to reflect performance when the assay is used in routine surveillance, that is, when there are no signs of renal injury. This cohort of 114 samples showed a 92.3% sensitivity (95% CI, 64.0–99.8%) and 75.2% specificity (95% CI, 65.7–83.3%) for detection of AR. The AUC was 0.89 (95% CI, 0.76–0.99). Based on a 25% prevalence of rejection in an at-risk population, the positive predictive value (PPV) was projected to be 55.4% (95% CI, 46.2–64.7%) and the negative predictive value (NPV) was projected to be 96.7% (95% CI, 90.6–99.9%).

Sensitivity, specificity, PPV and NPV were also calculated at different dd-cfDNA level rejection cutoffs. Appendix A shows the metrics at 0.6%, 0.8%, 1.0%, 1.2%, 1.4%, and 1.6%. Raising the cutoff has the effect of improving the specificity and the PPV; lowering the cutoff improves sensitivity and NPV.

### 3.5. Relationship Between dd-cfDNA and Rejection Type

Of the 38 samples with biopsy-proven AR, 16 were classified as either ABMR or ABMR and borderline T-cell-mediated rejection (bTCMR); 12 had a combination of both ABMR and TCMR; 10 were classified as either TCMR or TCMR and bABMR. In addition, 13 and 59 BL samples were classified as bAMBR and bTCMR, respectively. Figure 5 shows the relationship between dd-cfDNA level and type of rejection (for groups with known ABMR or known TCMR). Median dd-cfDNA did not differ significantly between AMBR (2.2%), ABMR/TCMR (2.6%), or TCMR (2.7%) groups (*p* = 0.855) (Appendix A). The study contained a range of pathologies, and the data indicate that this assay, unlike other published studies measuring cfDNA by other assays [14], is robust to different rejection types (Appendix A). The dd-cfDNA breakdown of bABMR and bTCMR samples are depicted in Appendix A. 

### 3.6. dd-cfDNA Levels by Donor Type

To assess the relationship between dd-cfDNA and donor type (living related, living non-related, and deceased non-related) a linear mixed-effects model was constructed using a log transformed dd-cfDNA as the response and donor type as the predictor for the non-rejection group. The log-transformation was applied to satisfy the model’s assumptions. The test was limited to the non-rejection group due to the limited number of AR samples in two groups (living related and living non-related). An ANOVA Wald-test with Kenward–Roger approximation for the degrees of freedom showed significance (*p* = 0.045). Tukey’s post-hoc test was used to determine the difference among the three groups: none of the post-hoc tests demonstrated any association (Figure 6, Appendix A). It is possible that the overall effect is driven by a sub-category of the non-rejection group (STA, BL, or OI) or the effect between the groups is smaller than detectable with the current sample size [30].

### 3.7. dd-cfDNA Variability over Time

Two analyses were designed to evaluate the natural variability in dd-cfDNA over time in biopsy-matched, non-rejection patients. The first sub-analysis was a cross-sectional analysis of 60 plasma samples from 60 different patients, collected immediately following surgery (within three days (“Day 0”)) or at 1, 3, 6, or 12 months post-surgery. Among these STA patients, dd-cfDNA levels were lower at month 0 than subsequent time points; however, for most of these STA samples dd-cfDNA levels were <1% across all time points (Figure 7A). No association was observed between Day 0 samples and the other time points, although the overall distribution of dd-cfDNA levels in the Day 0 group appears lower in comparison (Figure 7A). 

To evaluate the normal intra-patient variation in donor fraction, the second sub-analysis longitudinally assessed 10 individual patients across four time points (varying between about 1 month and 1 year post transplantation (minimum interval: 11 days, maximum interval: 345 days)). Overall, organ injury occurred at dd-cfDNA levels above 1% (Figure 7B). The inter-patient standard deviation within this cohort was 0.16 (95% CI, 0.0–0.37) and the intra-patient standard deviation was 0.42 (95% CI, 0.32–0.56). The intraclass-correlation coefficient was low (0.1193), which suggests that the variability in these data are mostly due to intra-patient variation. Figure 7C depicts all available longitudinal data among patients that experienced a rejection. In 9/11 patients, dd-cfDNA levels were above 1% prior to rejection.

## 4. Discussion

In this study, median dd-cfDNA was significantly higher in the AR group (2.32%) versus the non-rejection group (0.47%; *p* < 0.0001). Analysis of performance estimates demonstrated that the mmPCR-NGS method was able to discriminate active from non-rejection status with an AUC of 0.87 and high sensitivity (88.7%) and specificity (72.6%) at an AR cutoff of >1% dd-cfDNA. Based on a 25% prevalence of rejection, projected PPV and NPV were 52.0% and 95.1%, respectively. In contrast, eGFR scores were generally less discriminatory, with a 67.7% sensitivity and 65.3% specificity, and projected PPV and NPV of 39.4% and 85.9%, respectively. Therefore, if eGFR measurements were used as the sole clinical decision point, about 1 in 7 patients found to be at low risk of rejection would actually be experiencing rejection, and would not be referred for an indication biopsy—this is in comparison to the projected NPV for dd-cfDNA that suggests that only 1 in 20 patients would miss an indication biopsy where it might be clinically necessary. Taken together, the superior performance of this SNP-based dd-cfDNA assay over that of the current standard of care for the evaluation of allograft rejection holds promise for enabling patients a greater opportunity for timely therapy in the case of an allograft injury.

Levels of dd-cfDNA also provided discrimination of AR from the three non-rejection subgroups (STA, BL, and OI); median dd-cfDNA levels were significantly higher for samples with biopsy-proven AR (2.3%) versus BL (0.6%), OI (0.7%), and STA (0.4%). In a post hoc analysis, we examined the ability of dd-cfDNA combined with eGFR to predict rejection status (AR/non-rejection) in biopsy matched samples (Appendix A). This combined approach correctly classified 32/38 (84.2%) AR and 145/179 (81.0%) non-rejection samples, though in a head-to-head comparison it showed little to no improvement over dd-cfDNA alone. Combining dd-cfDNA with other markers may provide improved predictive value, but this was outside the scope of this study. Also of note, while both dd-cfDNA and eGFR can be used to differentiate AR and STA cases, the BL and OI samples stratify differently: they tend to aggregate with STA when using dd-cfDNA and with AR when using eGFR. This suggests that dd-cfDNA could be used together with eGFR to differentiate patients into three groups—STA patients, AR patients, and patients experiencing BL or OI. 

In a recent study that amplified hundreds of target SNPs in dd-cfDNA to detect active rejection in kidney allografts, that method was able to discriminate AR from non-rejection with an AUC of 0.74, 59% sensitivity, and 85% specificity [14]. In comparison with that study, the novel dd-cfDNA test described in the current study showed a higher AUC value (0.87) as well as greater sensitivity (89%). On the other hand, specificity (73%) was slightly lower in the current study, partly driven by the fact that a majority of the “false positives” were cases with BL and OI indicating some form of organ injury. The predefined analysis in this study used 1% dd-cfDNA cutoff, based on prior experience [14]; however, as a different sensitivity/specificity tradeoff may be optimal in different use cases, performance was calculated, in a post hoc fashion, for additional cfDNA cutoffs: 0.6%, 0.8%, 1.2%, 1.4%, and 1.6 % (Appendix A). 

Another important finding of this study was that the fraction of dd-cfDNA did not differ between ABMR and TCMR groups, with dd-cfDNA levels of 2.2% and 2.7%, respectively. These results are novel considering that a previously conducted study by Bloom et al. (2017), which used a different assay, found significantly higher dd-cfDNA levels for ABMR (2.9%) than for TCMR (≤1.2%) [14], showing a lower ability to detect T-cell mediated rejections. Though the assay used in that study also measured dd-cfDNA, the methods used by the two assays differ greatly. It is unclear whether that test could not differentiate AR from non-rejection in cases of TCMR or if the result was due to the smaller sample size of that group in that study (*n* = 11). Regardless, it appears that dd-cfDNA measurements based on the mmPCR assay in this study can accurately discriminate AR from non-rejection across a range of pathologies, including both acute and chronic findings, in both the ABMR and TCMR groups. An additional finding in this study is that borderline, or early rejection injury, has a lower burden of dd-cfDNA than more established injury, making it possible to use this sensitive assay to track evolution of, or recovery from, AR.

One barrier to widespread clinical use of dd-cfDNA as a diagnostic tool for monitoring organ transplant has been the limitations in measuring dd-cfDNA in certain cases, such as when the donor genotype is unknown or when the donor is a close relative. Given the design of the assay used here, it is possible to quantify dd-cfDNA without prior recipient or donor genotyping. Further, there is no need for a computational adjustment based on whether the donor is related to the recipient. In this study, evaluation of dd-cfDNA levels by donor type revealed that regardless of donor type (living related, living non-related, deceased non-related), dd-cfDNA levels were similar across all donor types within in the AR and non-rejection categories. 

A limitation of this study is that it was a retrospective analysis of archived samples from a single center. However, the central geographical area enabled all biopsies to be performed by a single pathologist, which may have helped minimize variability in biopsy classification; further, all experimenters were kept blinded during the process of data generation. The retrospective study design may have led to differences in patient characteristics across the rejection groups; for example, the STA group was enriched with younger patients who may be better suited immunologically to tolerate transplanted organs compared to older-aged patients. However, these age differences likely did not affect the validity of the study findings.

A strength of this study is the large number of samples drawn at the time of a protocol biopsy. Performance of the assay among samples drawn at the time of a protocol biopsy are more reflective of expected performance during routine use of the assay, where there are no overt signs of injury; this is in contrast to for-cause biopsies, which are performed in a high-risk cohort where there are peripheral signs of organ injury. In this study, more than half (53%, 114/217) of the biopsy-matched samples were performed on protocol. The assay showed better performance in this cohort, with a sensitivity of 92.3%, specificity of 75.2%, and AUC of 0.89%. This data suggests that application of the dd-cfDNA assay in a clinical setting could potentially reduce the need for protocol biopsies.

Another strength is the variety of patient samples in the non-rejection group, which comprised not only STA, but also BL and OI samples. This allowed for additional analyses in this study, which found that dd-cfDNA was significantly different in the AR group versus BL and OI groups. Additional sub-analyses by type of AR (ABMR and TCMR), as well as by donor type, demonstrated that dd-cfDNA levels were able to discriminate AR versus non-rejection in a variety of rejection and patient types. Further, the SNP-based mmPCR methodology underlying this assay has been extensively validated in the context of prenatal testing, and has been used to determine the DNA fraction of the minor constituent in a clinical setting in over a million maternal/fetal DNA samples. Finally, the inclusion of longitudinal data enabled a unique evaluation of the natural variability of dd-cfDNA in transplant patients over time. Inter-patient variability data demonstrated that between 0 and 12 months post-surgery, most patients with STA biopsies had dd-cfDNA levels below 1%, and most patients with a positive biopsy had a positive dd-cfDNA test at a time point prior to the positive biopsy. Taken together, this suggests that this mmPCR assay may be used for routine monitoring, to determine whether a renal transplant patient is experiencing organ injury that may require a change in management.

## 5. Conclusions

In conclusion, this study validates the use of dd-cfDNA in the blood as an accurate marker of kidney injury/rejection across a range of pathologies with acute and chronic findings. This rapid, accurate, and noninvasive technology allows for detection of significant renal injury in patients better than the current standard of care, with the potential for better patient management, more targeted biopsies, and improved renal allograft function and survival. 

## Figures and Tables

**Figure 1 jcm-08-00019-f001:**
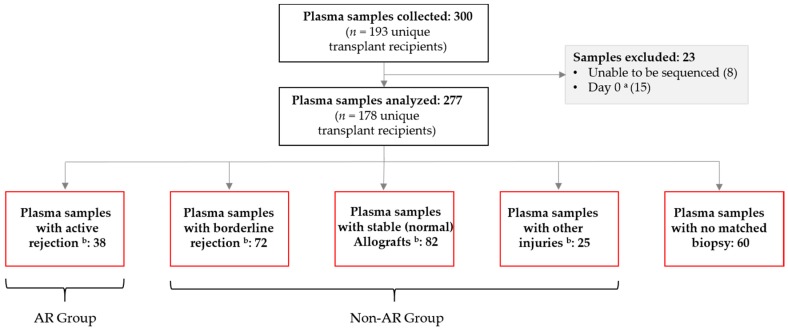
Plasma sample breakdown. AR, active rejection. ^a^ Collected within three days post transplantation; ^b^ samples drawn on the day of biopsy (i.e., were biopsy-matched).

**Figure 2 jcm-08-00019-f002:**
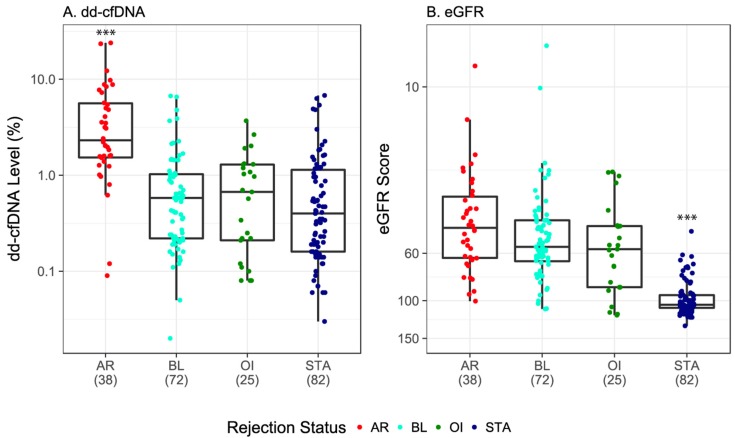
Discrimination of active rejection by dd-cfDNA versus eGFR. (**A**) and (**B**): Boxes indicate interquartile range (25th to 75th percentile); horizontal lines in boxes represent medians; each dot depicts one sample. *p*-values for dd-cfDNA and eGFR adjusted using Kruskal–Wallis rank sum test followed by Dunn multiple comparison tests with Holm correction. *** indicates adj. *p* < 0.0001 from all other group comparisons (see Appendix A). AR, active rejection; BL, borderline; OI, other injury; STA, stable; dd-cfDNA, donor-derived cell-free DNA; eGFR, estimate glomerular filtration rate.

**Figure 3 jcm-08-00019-f003:**
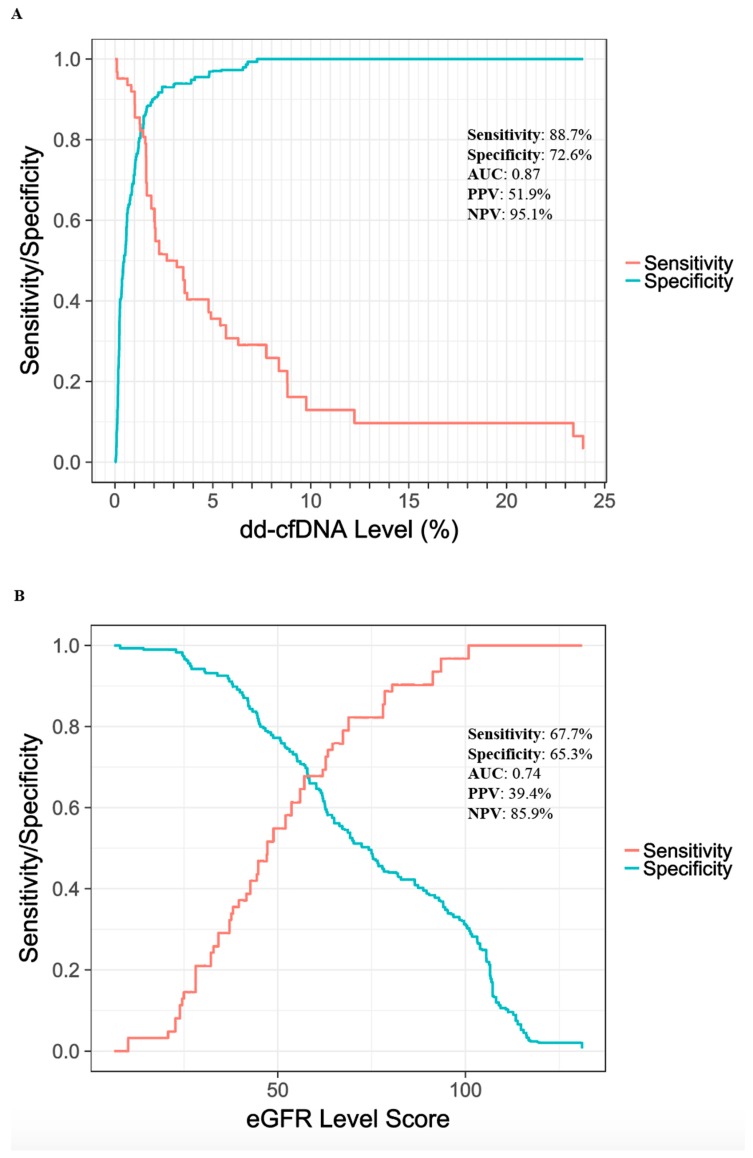
Predictive statistics for active rejection versus non-rejection. Sensitivity (red line) and specificity (blue line) are depicted over the observed range of dd-cfDNA levels (**A**) and eGFR scores (**B**). Reported sensitivity and specificity correspond to cutoffs of 1% for dd-cfDNA and a score of 60 for eGFR. PPV and NPV are based on a 25% AR prevalence. AUC, area under the curve; PPV, positive predictive value; NPV, negative predictive value.

**Figure 4 jcm-08-00019-f004:**
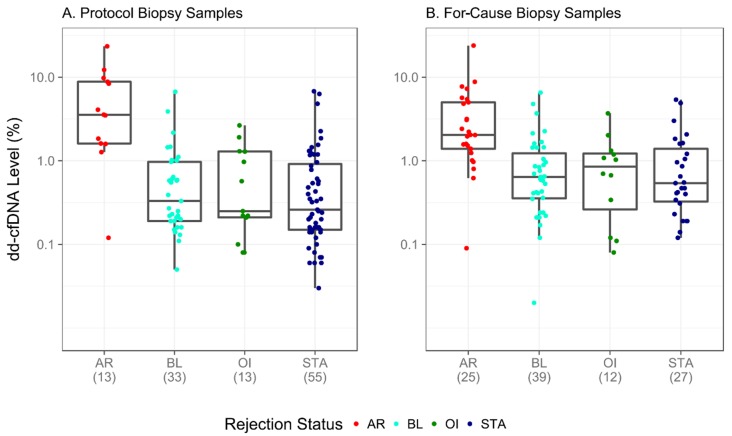
Discrimination of active rejection by dd-cfDNA in biopsy-matched samples stratified by biopsy type. The number of samples per group and the distribution of their dd-cfDNA levels are depicted for protocol biopsy (**A**) and for-cause biopsy (**B**) samples. Boxes indicate inter-quartile range, horizontal lines represent medians. AR, active rejection; BL, borderline; OI, other injury; STA, stable.

**Figure 5 jcm-08-00019-f005:**
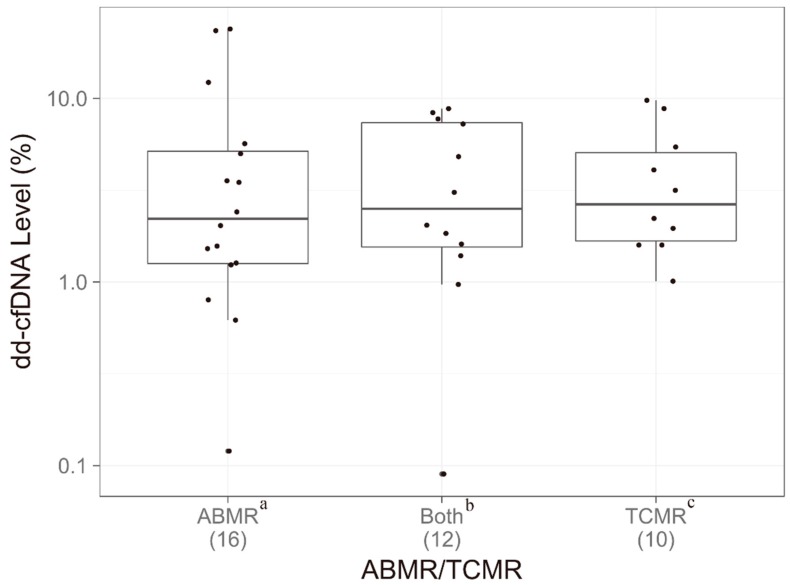
dd-cfDNA as a function of antibody-mediated—versus T-cell—mediated rejection. Boxes indicate interquartile range (25th to 75th percentile); horizontal lines in boxes represent medians; dots indicate all individual data points. *p*-values for dd-cfDNA adjusted using Kruskal–Wallis rank sum test. ^a^ Samples assigned ABMR or ABMR and bTCMR. ^b^ Samples assigned ABMR and TCMR. ^c^ Samples assigned TCMR or TCMR and bABMR. ABMR, antibody-mediated rejection; TCMR, T-cell-mediated rejection.

**Figure 6 jcm-08-00019-f006:**
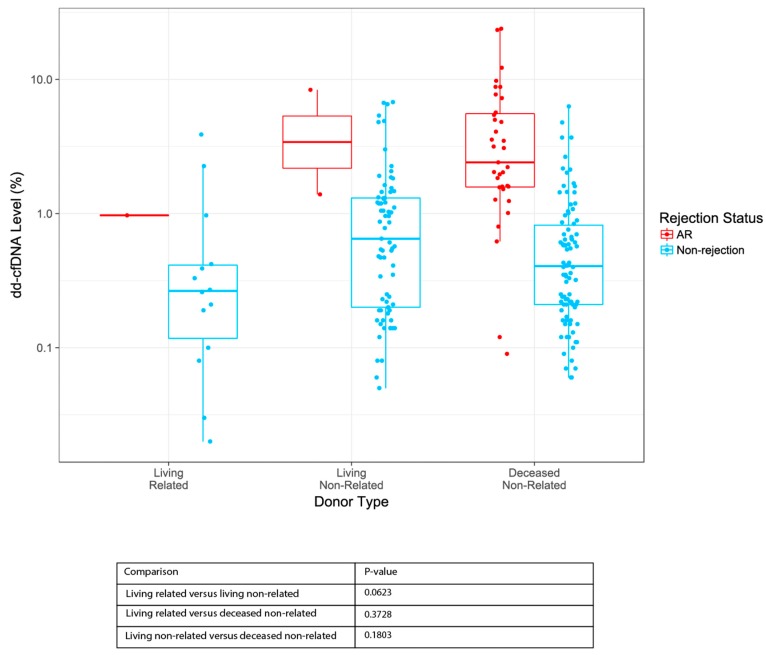
Relationship between dd-cfDNA and donor type. Boxes indicate inter-quartile range, horizontal lines represent medians. *p*-values for dd-cfDNA an ANOVA Wald-test with Kenward–Roger approximation for the degrees of freedom was followed by Tukey’s post-hoc test. AR, active rejection.

**Figure 7 jcm-08-00019-f007:**
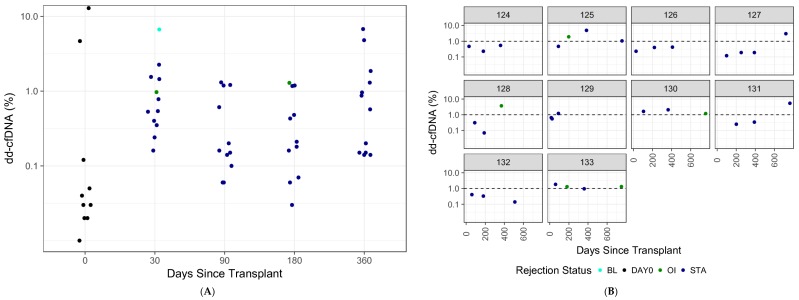
Variability in dd-cfDNA in non-rejection patients over time. (**A**) Inter-patient variability (60 samples from 60 patients over time); (**B**) intra-patient variability (samples from the same 10 patients over time); (**C**) change in dd-cfDNA levels over time in patients with active rejection. AR, active rejection; BL, borderline; OI, other injury; STA, stable.

**Table 1 jcm-08-00019-t001:** Demographics and characteristics ^a^.

Phenotype Characteristic	Active Rejection (38 Samples)	Non-Rejection			
		Stable (82 Samples)	Borderline AR (72 Samples)	Other Injury (25 Samples) ^b^	Combined (179 Samples)
Recipient age, year ***** (*p*-value < 0.0001)					
(0, 18)	0 (0)	44 (53.7)	1 (1.4)	4 (16.0)	49 (27.4)
(18, 40)	10 (26.3)	32 (39.0)	18 (18.0)	8 (32.0)	58 (32.4)
(40, 80)	28 (73.7)	6 (7.3)	53 (73.6)	13 (52.0)	72 (40.2)
Mean ± SD	47.91 ± 14.31	20.04 ± 11.97	47.88 ± 13.24	44.75 ± 23.73	34.65 ± 19.87
Median	49.13	19.96	47.46	40.97	31.33
Range	23–76	3–70	5–74	3–80	3–80
Male/female, no. (%) (*p*-value = 0.5988)					
Male	17 (44.7)	48 (58.5)	40 (55.6)	15 (60)	103 (57.5)
Female	21 (55.3)	34 (41.5)	32 (44.4)	10 (40)	76 (42.5)
Ethnicity, no. (%) (*p*-value = 1)					
Hispanic or Latino	13 (34.2)	28 (34.1)	24 (33.3)	10 (40)	62 (34.6)
Not Hispanic or Latino	25 (65.8)	54 (65.9)	48 (66.7)	15 (60)	117 (65.4)
Race groups, no. (%) (*p*-value = 0.4695)					
White or Caucasian	10 (26.6)	42 (51.2)	16 (22.2)	6 (24)	64 (35.8)
Black or African American	6 (15.8)	7 (8.5)	14 (19.4)	4 (16)	25 (14.0)
Asian or Pacific Islander	8 (21.1)	4 (4.9)	15 (20.8)	4 (16)	23 (12.8)
Other/Not reported	14 (36.8)	29 (35.4)	27 (37.8)	11 (44.0)	67 (37.4)
Recipient weight, kg (*p*-value = 0.6039)					
Mean ± SD	76.22 ± 19.7	70.9 ± 8.8	79.18 ± 18.7	78.33 ± 17.1	78.1 ± 17.6
Median	72.5	73.0	78.0	76.0	76.0
Range	45–119	52–81	46–134	47–109	46–134
Unknown	6	72	7	7	86
DSA positive, no. (%) (*p*-value = 0.1928)					
Yes	15 (39.5)	0 (0)	18 (25)	2 (8)	20 (11.2)
No	21 (55.3)	0 (0)	48 (66.7)	3 (12)	51 (28.5)
Not recorded	2 (5.3)	82 (100)	6 (8.3)	20 (80)	108 (60.3)
Indication for renal transplantation, no. (%) (*p*-value = 0.4869)					
Glomerulonephritis	5 (13.2)	6 (7.3)	4 (5.6)	1 (4)	11 (6.1)
Focal segmental glomerulosclerosis	5 (13.2)	5 (6.1)	6 (8.3)	2 (8)	13 (7.3)
Diabetes mellitus	5 (13.2)	3 (3.7)	15 (20.8)	5 (20)	23 (12.8)
Thin basement membrane nephropathy	0 (0)	0 (0)	2 (2.8)	0 (0)	2 (1.1)
Polycystic kidney disease	3 (7.9)	2 (2.4)	7 (9.7)	1 (4)	10 (5.6)
Solitary kidney	0 (0)	0 (0)	3 (4.2)	0 (0)	3 (1.7)
Hypertension	4 (10.5)	2 (2.4)	13 (18.1)	3 (12)	18 (10.1)
IgA nephropathy	3 (7.9)	0 (0)	7 (9.7)	1 (4)	8 (4.5)
Lupus nephritis	2 (5.3)	0 (0)	0 (0)	0 (0)	0 (0.0)
ANCA—vasculitis	1 (2.6)	0 (0)	2 (2.8)	0 (0)	2 (1.1)
Other/Unknown	10 (26.3)	64 (78.1)	13 (18.1)	12 (48)	89 (49.7)
Donor source *, no. (%) (*p*-value < 0.0001)					
Living related	1 (2.8)	2 (2.4)	9 (12.5)	3 (12)	14 (7.8)
Living unrelated	2 (5.3)	50 (61)	18 (25)	7 (28)	75 (41.9)
Deceased unrelated	35 (92.1)	30 (36.6)	45 (62.5)	15 (60)	90 (50.3)

* Indicates the association with AR status (AR/non rejection) was statistically significant (*p* < 0.001). Categorical variables were tested using Fisher’s exact test for count data, and numerical variables were tested using a likelihood ratio test based on a logistic regression. ^a^ Characteristics and demographic information are based on all samples drawn on the day of biopsy; data reflects multiple samples for some patients. ^b^ Other injuries included: chronic allograft nephropathy (10 samples), drug toxicity (11 samples), BK nephritis (1 sample), acute tubular necrosis (1 sample), transplant glomerulopathy (1 sample), and post borderline-TCMR (1 sample). DSA, donor specific antibodies; AR, active rejection.

**Table 2 jcm-08-00019-t002:** Summary statistics for donor-derived cell-free DNA (dd-cfDNA) and estimated glomerular filtration rate (eGFR) variables across AR and non-rejection groups.

Parameter	Active Rejection	Non-Rejection			
		Stable	Borderline AR	Other Injury	Combined
dd-cfDNA					
Number of samples (%)	38 (17.5)	82 (37.8)	72 (33.2)	25 (11.5)	179 (82.5)
Mean (SD)	4.64 (5.45)	0.90 (1.36)	0.95 (1.31)	0.89 (0.91)	0.92 (1.28)
Median (range)	2.32 (0.1–23.9)	0.4 (0.03–6.8)	0.58 (0.02–6.7)	0.67 (0.08–3.69)	0.47 (0.04–6.78)
eGFR					
Number of samples (%)	38 (17.5)	82 (37.8)	72 (33.2)	25 (11.5)	179 (82.5)
Score mean (SD)	49.0 (22.4)	99.5 (16.1)	55.9 (21.4)	63.8 (29.0)	77.0 (8.45)
Score median (range)	45.67 (8.0–100.4)	104.5 (47.4–131.1)	55.99 (6.4–109.4)	57.4 (25.0–116.9)	76.06 (6.4–131.1)

AR, active rejection.

**Table 3 jcm-08-00019-t003:** Cohort breakdown into for-cause and protocol biopsy.

Rejection Status	Biopsy Reason	Total	Median	Low	High	Mean	SD
AR	For-cause	25	2.04	0.09	23.9	3.85	4.81
	Protocol	13	3.56	0.12	23.4	6.16	6.44
BL	For-cause	39	0.64	0.02	6.54	1.07	1.32
	Protocol	33	0.33	0.05	6.69	0.82	1.30
OI	For-cause	12	0.865	0.08	3.69	1.03	1.02
	Protocol	13	0.25	0.08	2.65	0.76	0.82
STA	For-cause	27	0.54	0.12	5.38	1.12	1.36
	Protocol	55	0.26	0.03	6.78	0.80	1.37

AR, active rejection; BL, borderline; OI, other injury; STA, stable.

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
