# Peer review of "Optimizing Detection of Kidney Transplant Injury by Assessment of Donor-Derived Cell-Free DNA via Massively Multiplex PCR"

_jcm, 2018, doi:10.3390/jcm8010019_

Round 1

Reviewer 1 Report

Title: Optimizing Detection of Kidney Transplant Injury by Assessment of Donor-Derived Cell-Free DNA via Massively Multiplex PCR

This paper presents that a novel single nucleotide polymorphism (SNP)-based massively multiplexed PCR (mmPCR) methodology to measure dd-cfDNA in various types of renal transplant recipients for the detection of allograft rejection/injury without prior knowledge of donor genotypes. The authors showed that the use of dd-cfDNA in the blood as an accurate marker of kidney injury or rejection across a range of pathologies with accurate and chronic findings. Below are my comments for the authors to address.

Section 2.5: Statistical Analyses

o   Please do the citation for the methods were applied in the manuscript that can help reader to follow easily. As an example, the Holm approach, the non-parametric multiple comparison test Dunn need to be cited, or even the bootstrap approach in computing the ROC cruve.  

o   The authors considered the Holm approach to adjust for multiple hypothesis testing. I was wondering to know why the authors considered the Holm correction instead of considering Benjamini-Hochberg (BH), FDR, and Bonferroni (depends on the number of tests) which is more common in adjusting p-values for the multiple comparisons.

o   In the AUC analysis, the authors did the bootstrap method or the Delong method? Did the authors use the Delong method to compare the areas under the curves? Please clarify this part.

o   The authors used the log-transformation to fit the mixed effects model. I guess they used the linear mixed effect models. Please clarify this part and cite to the paper.

§  Why the authors didn’t use log-transformation for the dd-cfDNA and use the parametric test instead of non-parametric test Kruskal-Wallis (KW) test?

2.      Section 3.1: Results

o   Authors did the some intervals by words for the age variable, less than 18 yrs, between 18 and 40 yrs, and greater than 40 yrs. If the authors think this category is important why not to be included in the Table 1. Instead of having the continuous age variable, authors can have the categorical variable for age variable.

o   Authors mentioned that “The data shown that the dd-cfDNA levels in patients with AR is significantly higher than patients with STA, BL, or OL”. Could you please clarify this sentence and cite which data or results can clarify this sentence?

o   In Table 1 please add the “median” for the continuous variables, and keep (%) for all categorical variables (e.g., Ethnicity).

o   Please add the p-values to test whether these variable in Table 1 are associated with rejection group (i.e., AR and non-rejection). Please add the statistical method that authors will use (e.g., Fisher exact test for categorical data with few number of samples) in the statistical method section.  Please also mark those significant p-values in the Table 1 (e.g., p-values<0.05).

3.      Section 3.2: Results

o   The authors explained in statistical method section (2.5) that the non-parametric method KW test along with Dunn multiple comparison tests with Holm is used. While later in section 3.2, the Tukey’s test was used for eGFR data (Caption Fig. 2C). Please clarify this part and keep consistency all over the manuscript. If you changed the method of analysis for eGFR please explain briefly the reason of that. And, also add in the statistical method section (2.5).

o   Please consider Fig. 2C as a Table and before Fig. 2(A-B), since it represents the descriptive statistics for the variables dd-cfDNA and eGFR across AR and non-rejection groups.

§ For Fig. 2C, please keep (%) and score in front of the variable name (e.g., eGFR (score) and dd-cfDNA (%)). Also, please add the percentage in parentheses in front of all frequency values for both dd-cfDNA and eGFR varibales (e.g., 38 (a%)).

§ Authors talked about the comparison between AR and non-rejection group for both eGFR and dd-cfDNA variables in the manuscript. Please add those p-values in the Fig. 2C. If the authors did the non-parametric KW method, that’s fine, otherwise please explain in the method’s section. As an example, the first two lines in section 3.2 represent the comparison between AR and non-rejection for dd-cfDNA which is not reported anywhere. Then, please add those p-values in Fig. 2C.

o   Recommend the authors to do the citations more precisely for Figs and Tables. For example, Fig. 2A (results show for the dd-cfDNA) or Fig. 2B (results show for the eGFR).

o   Last sentences in paragraph one (page 6), authors said “there was no significant difference in the level of dd-cfDNA between any of the non-rejection subgroup …”. Please add the Dunn results (both Statistics and adjusted p-value) as a Suppl. Table to clarify this part. Please apply this comment for eGFR variable.

o   Please modify Fig. 2A by considering the lower value for the y-axis (the points are not clear for STA group). Please double check the Fig. 2A. The Range shows the BL has the min value (0.02), while the values for STA group (with min 0.08) are smaller!

o   Please add the Suppl. Fig to show the Boxplot for dd-cfDNA variable based on two groups (AR and non-rejection). Do the same thing for the eGFR variable.

4.      Section 3.4: Results

o   Please change Fig. 4C to the Table. Then, add more descriptive statistics to that like Fig. 2C. (e.g., replace the dd-cfDNA and eGFR by Clinical Reason and Protocol).

o   Move the Fig. 4C before Fig.4 (A-B).

o   The authors demonstrate the results by words for the dd-cfDNA levels > 1%, while the Fig. 4 shows even less than 1%. It is not easy to be followed!

o   Increasing the cutoff level of rejection may improve the specificity and PPV. Why the authors consider the cutoff levels between 1%-1.6%? And the increases in the cutoff levels can reduce the number of samples under each group, and how this reduction can change the power of analysis? Please clarify this part.  

o   Table S1: please also add the ROC curve (similar to Fig. 3) for all cutoff points as Suppl. Fig.

5.      Section 3.5: Results

o   Please remove the title of Fig. 5.

o   Since the non-parametric KW test is not significant, then why the Dunn test was used?

o   Provide the Suppl. Table to present the descriptive statistics for this Fig. 5 (similar to the Fig. 2C or Fig. 4C).

6.      Section 3.6: Results

o   Please add the Suppl. Table for this part (similar to the Fig. 2C or Fig. 4C).

o   Taking the mean for duplicate data is not always a good option. I would like to ask the authors to track the duplicate samples in the Fig. 6 by using different colors.

o   Also, how about fitting the mixed models if we have enough duplicates? If there are enough duplicates in each group, recommend the authors to fit the mix effects model to see if the results can catch any differences between groups.

o    “No significance difference by donor type was observed”. Does it mean between any of the non-rejection group and donor type (p>0.46)? If yes, why the authors again considered the Dunn test?   

7.      In Discussion section:  

o   Page 16 lines 356-357: Authors explain that “age differences likely did not affect the validity of the study findings”.  Did authors do any test to observe that? Please clarify this sentence.

8.      General comments for authors:

o   It seems that there are some duplicates values for the patients (e.g., Fig. 6). If there are any duplicates for other datasets, please specify precisely for the duplicates across the manuscript and if it is possible try to show that in the Figs by using different colors to emphasize that. 

o   Since this paper is not the statistical paper, then instead of using only word “significant” based on p-values (or adjusted p-values), please explain by words that as an example there is not any association between A and B (easy for reader to follow). Also, the authors used “p” to demonstrate both the p-values and adjusted p-values. Please specify more precisely across the manuscript.

Author Response

Dear Reviewer 1, Thank you for your review and please see the point-by-point response in the uploaded PDF.

1. SECTION 2.5: STATISTICAL ANALYSES
Point 1. Please do the citation for the methods were applied in the manuscript that can help reader to follow easily. As an example, the Holm approach, the non-parametric multiple comparison test Dunn need to be cited, or even the bootstrap approach in computing the ROC cruve.
Response 1: Thank you - the following citations for the Dunn test, Holm correction, and Bootstrap method were added to the text: • Dunn, O.J. 1964. Multiple comparisons using rank sums. Technometrics 6:241-252. • Holm, S. (1979). "A simple sequentially rejective multiple test procedure". Scandinavian Journal of Statistics. 6 (2): 65- 70 • Efron, B.; Tibshirani, R. Bootstrap Methods for Standard Errors, Confidence Intervals, and Other Measures of Statistical Accuracy. Statist. Sci. 1 (1986), no. 1, 54--75. doi:10.1214/ss/1177013815. https://projecteuclid.org/euclid.ss/1177013815
Point 2: The authors considered the Holm approach to adjust for multiple hypothesis testing. I was wondering to know why the authors considered the Holm correction instead of considering Benjamini-Hochberg (BH), FDR, and Bonferroni (depends on the number of tests) which is more common in adjusting p-values for the multiple comparisons.
Response 2: Holm was chosen for the ability of controlling the overall type I error. Holm is an extension to the Bonferroni correction that gives a more powerful adjustment.
Point 3. In the AUC analysis, the authors did the bootstrap method or the Delong method? Did the authors use the Delong method to compare the areas under the curves? Please clarify this part.
Response 3: The DeLong method was used to estimate the appropriate standard error of the AUC value. We can compare the AUCs of dd-cfDNA and eGFR with a bootstrap procedure (the ROC curves have different directions, so DeLong might not work well). This comparison is not significative (p-value=0.117).
Point 4. The authors used the log-transformation to fit the mixed effects model. I guess they used the linear mixed effect models. Please clarify this part and cite to the paper.
Response 4: The model used is a Linear Mixed-Effects Model.
The following citation for the Linear Mixed-Effects Model approach was added to the text:
• Laird, Nan M., and James H. Ware. “Random-Effects Models for Longitudinal Data.” Biometrics, vol. 38, no. 4, 1982, pp. 963–974. JSTOR, JSTOR, www.jstor.org/stable/2529876.
Point 5. Why the authors didn’t use log-transformation for the dd-cfDNA and use the parametric test instead of non-parametric test Kruskal-Wallis (KW) test?
Response 5: The non-parametric test was chosen in most of the tests for its interpretability. A significant result can be interpreted as a difference in the median of the distributions. When looking at longitudinal data if we want to consider the patient effect, a mixed-model is a straightforward way of doing this, but this requires that you have approximately normal data, hence the need of a transformation. It would be possible to use ranks instead of a transformed response but that seems to increase the complexity of the model. Moreover, the results we obtain with a transformation and a parametric test are not very different.
2. SECTION 3.1: RESULTS
Point 6. Authors did the some intervals by words for the age variable, less than 18 yrs, between 18 and 40 yrs, and greater than 40 yrs. If the authors think this category is important why not to be included in the Table 1. Instead of having the continuous age variable, authors can have the categorical variable for age variable.
Response 6: Thank you – this was added to Table 1.
Point 7. Authors mentioned that “The data shown that the dd-cfDNA levels in patients with AR is significantly higher than patients with STA, BL, or OL”. Could you please clarify this sentence and cite which data or results can clarify this sentence?
Response 7. The quoted text above only was present on lines 164-165, citing a reference (not referring to this study’s data): “Published data have shown that the dd-cfDNA levels in patients with AR is significantly higher than patients with STA, BL, or OI.24” To make this more clear, we inserted a paragraph return at the start of the sentence in question, and clarified the data presented in reference 24.
Point 8. In Table 1 please add the “median” for the continuous variables and keep (%) for all categorical variables (e.g., Ethnicity).
Response 8: This edit was made in Table 1, thank you.
Point 9. Please add the p-values to test whether these variable in Table 1 are associated with rejection group (i.e., AR and non-rejection). Please add the statistical method that authors will use (e.g., Fisher exact test for categorical data with few number of samples) in the statistical method section. Please also mark those significant p-values in the Table 1 (e.g., p-values<0.05).
Response 9: For categorical data a Fisher Exact test was used, and for continuous variables the significance was tested by using a likelihood ratio tested based on a logistic regression of the variable and the rejection status. Missing values were removed before applying the tests.
3. SECTION 3.2: RESULTS
Point 10. The authors explained in statistical method section (2.5) that the non-parametric method KW test along with Dunn multiple comparison tests with Holm is used. While later in section 3.2, the Tukey’s test was used for eGFR data (Caption Fig. 2C). Please clarify this part and keep consistency all over the manuscript. If you changed the method of analysis for eGFR please explain briefly the reason of that. And, also add in the statistical method section (2.5).
Response 10: The legend had incorrect information in it and has been corrected. Thank you.
Point 11. Please consider Fig. 2C as a Table and before Fig. 2(A-B), since it represents the descriptive statistics for the variables dd-cfDNA and eGFR across AR and non-rejection groups.
For Fig. 2C, please keep (%) and score in front of the variable name (e.g., eGFR (score) and dd-cfDNA (%)). Also, please add the percentage in parentheses in front of all frequency values for both dd-cfDNA and eGFR varibales (e.g., 38 (a%)).
Authors talked about the comparison between AR and non-rejection group for both eGFR and dd-cfDNA variables in the manuscript. Please add those p-values in the Fig. 2C. If the authors did the non-parametric KW method, that’s fine, otherwise please explain in the method’s section. As an example, the first two lines in section 3.2 represent the comparison between AR and non-rejection for dd-cfDNA which is not reported anywhere. Then, please add those p-values in Fig. 2C.
Response 11: The original 2C (Summary statistics) information was removed from Fig 2 and is now Table 2. For Figure 2, the significant comparisons are now indicated by a *** in Figure 2A and 2B, and the legend is updated to reflect this change. A multiple groups comparison table was also added to the Supplement (see Response 13).

Point 12. Recommend the authors to do the citations more precisely for Figs and Tables. For example, Fig. 2A (results show for the dd-cfDNA) or Fig. 2B (results show for the eGFR).
Response 12: This section was edited to refer to the tables and figure panels more clearly.
Point 13. Last sentences in paragraph one (page 6), authors said “there was no significant difference in the level of dd-cfDNA between any of the non-rejection subgroup …”. Please add the Dunn results (both Statistics and adjusted p-value) as a Suppl. Table to clarify this part. Please apply this comment for eGFR variable.
Response 13: The following supplemental table was added.

Point 14. Please modify Fig. 2A by considering the lower value for the y-axis (the points are not clear for STA group). Please double check the Fig. 2A. The Range shows the BL has the min value (0.02), while the values for STA group (with min 0.08) are smaller!
Response 14: The plot was updated and added to the manuscript:

Point 15. Please add the Suppl. Fig to show the Boxplot for dd-cfDNA variable based on two groups (AR and non-rejection). Do the same thing for the eGFR variable.
Response 15: The following graph was added to the supplement:

4. SECTION 3.4: RESULTS
Point 16. Please change Fig. 4C to the Table. Then, add more descriptive statistics to that like Fig. 2C. (e.g., replace the dd-cfDNA and eGFR by Clinical Reason and Protocol).
Response 16: The following tables have been added to the manuscript:

Point 17. Move the Fig. 4C before Fig.4 (A-B).
Response 17: The tables have been moved.
Point 18. The authors demonstrate the results by words for the dd-cfDNA levels > 1%, while the Fig. 4 shows even less than 1%. It is not easy to be followed!
Response 18: The section was edited to reduce unnecessary fraction when percentages were also shown. It now reads as follows:
“Among the biopsy-matched samples, 103 (47.5%) were biopsied for clinical reason, whereas 114 (52.5%) were biopsied according to protocol (Table 3; Table S3). Fig 4 depicts sample dd-cfDNA levels among all subgroups; 85 (39.2%) had dd-cfDNA levels >1%. Of those, 22 (25.9%) were STA; the remainder were AR (33 [38.8%]), OI (10 [11.8%]), or BL (20 [23.5%]). Of the individual groups, 33 (86.8%) of the total AR samples and 22 (26.8%) of the total STA samples had dd-cfDNA levels above 1%. In comparison, 20 (27.8%) of the total BL samples and 10 (40.0%) of the total OI samples had dd-cfDNA levels above 1%.”
Point 19. Increasing the cutoff level of rejection may improve the specificity and PPV. Why the authors consider the cutoff levels between 1%-1.6%? And the increases in the cutoff levels can reduce the number of samples under each group, and how this reduction can change the power of analysis? Please clarify this part.
Response 19: We have expanded our analysis to include cut off levels between 0.6% and 1.6% to see change in specificity and PPV. The number of samples in each group does not change by raising the cutoff since the group labels were given by biopsy so all the confidence intervals have the same “n” across the different scenarios. This is an exploratory look at the data and the impact to the certainty of the estimates is seen in the change in size of the confidence intervals.
Point 20. Table S1: please also add the ROC curve (similar to Fig. 3) for all cutoff points as Suppl. Fig.
Response 20: The plot corresponding to the different cutoffs would not be different than the one presented in figure 3. By changing the cutoff we select a different value of the x-axis (dd-cfDNA) and see what is the corresponding sensitivity and specificity (that are used in the PPV and NPV calculation). The AUC value stays constant.
5. SECTION 3.5: RESULTS
Point 21. Please remove the title of Fig. 5.
Response 21. The text has been removed and the figure updated:

Point 22. Since the non-parametric KW test is not significant, then why the Dunn test was used?
Response 22: There is no need to do the Dunn-Test, as there is no p-value for this, just the KW p-value. The legend for the figure has been updated:
Boxes indicate interquartile range (25th to 75th percentile); horizontal lines in boxes represent medians; dots indicate all individual data points. P-values for dd-cfDNA using Kruskal-Wallis rank sum test. a Samples assigned ABMR and bTCMR. bSamples assigned ABMR and TCMR. cSamples assigned TCMR and bABMR. ABMR, antibody-mediated rejection; TCMR, T-cell-mediated rejection.
Point 23. Provide the Suppl. Table to present the descriptive statistics for this Fig. 5 (similar to the Fig. 2C or Fig. 4C).
Response 23: The following table has been added to the manuscript:

6. SECTION 3.6: RESULTS
Point 24. Please add the Suppl. Table for this part (similar to the Fig. 2C or Fig. 4C).
Response 24: The following table has been added to the manuscript:

Point 25. Taking the mean for duplicate data is not always a good option. I would like to ask the authors to track the duplicate samples in the Fig. 6 by using different colors.
Response 25: We have done this; see the Figure pasted below. However, this figure is visually very difficult to interpret and we feel does not add clarity to the figure, therefore we did not include in the manuscript.

Point 26. Also, how about fitting the mixed models if we have enough duplicates? If there are enough duplicates in each group, recommend the authors to fit the mix effects model to see if the results can catch any differences between groups.
Response 26. This is a good point, we have changed the analysis to use a linear mixed model as this should handle the repeated measurements better. The test detects a difference among the groups but the pairwise comparisons turn out to be not significant. This could be due to having not enough samples in one of the categories or a small effect. The text of the section was changed to:
“To assess the relationship between dd-cfDNA and donor type (living related, living non-related, and deceased non-related) a linear mixed-effects model was constructed using a log transformed dd-cfDNA as the response and donor type as the predictor for the non-rejection group. The log-transformation was applied to satisfy the model’s assumptions. The test was limited to the non-rejection group due to the limited number of AR samples in two groups (living related and living non-related). An ANOVA Wald-test with Kenward-Roger approximation for the degrees of freedom showed significance (P=0.045). When performing Tukey’s post-hoc test was used to determine the difference among the three groups: none of the post-hoc tests demonstrated there was any association (Fig 6). It is possible that the overall effect is driven by a sub-category of the non-rejection group (STA, BL, or OI) or the effect between the groups is smaller than detectable with the current sample size.”1
1Kenward, M. G., and Roger, J. H. (1997). “Small Sample Inference for Fixed Effects from Restricted Maximum Likelihood.” Biometrics 53:983

The following reference to Kenward-Roger approximation was also added: Kenward, M. G., and Roger, J. H. (1997). “Small Sample Inference for Fixed Effects from Restricted Maximum Likelihood.” Biometrics 53:983–997.
Point 27. “No significance difference by donor type was observed”. Does it mean between any of the non-rejection group and donor type (p>0.46)? If yes, why the authors again considered the Dunn test?
Response 27: The Dunn test is not necessary, and it was added for completeness. Note that this has been removed since the analysis was changed to use a linear mixed model.
7. IN DISCUSSION SECTION:
Point 28. Page 16 lines 356-357: Authors explain that “age differences likely did not affect the validity of the study findings”. Did authors do any test to observe that? Please clarify this sentence.

Response 28: One test that we can do is see if the distribution of dd-cfDNA depends on age.

We can test for age after adjusting for rejection status (AR/Non-AR) with a linear mixed-effect model and an ANOVA Wald test with Kenward-Roger approximation for the degrees of freedom. In this case the test is not significative (P=0.48). The lack of AR samples in patients under 18 makes it harder to do a full comparison, but with the available data we cannot conclude that the outcome of a dd-cfDNA test would be much different for the different age groups. We will keep this in mind for future studies.
8. GENERAL COMMENTS FOR AUTHORS:
Point 29. It seems that there are some duplicates values for the patients (e.g., Fig. 6). If there are any duplicates for other datasets, please specify precisely for the duplicates across the manuscript and if it is possible try to show that in the Figs by using different colors to emphasize that.
Response 29: We have tried to do this by more colors or markers to the plots, however, it did not introduce more clarity, and instead made figures more confusing. See, e.g. our response to Figure 25, above. The patients with multiple samples are a minority in the data set, and had minimal impact on the conculsions.
Point 30. Since this paper is not the statistical paper, then instead of using only word “significant” based on p-values (or adjusted p-values), please explain by words that as an example there is not any association between A and B (easy for reader to follow). Also, the authors used “p” to demonstrate both the p-values and adjusted p-values. Please specify more precisely across the manuscript.

Response 30: Thank you for the feedback – the use of the word “significant” was reduced as suggested. The manuscript was also edited to indicate where adjusted p-values were used.

Reviewer 2 Report

In the present paper Sigdel et al. described a method based on the analysis of donor-derived cell-free DNA by massive multiplex PCR able to detect Kidney Transplantation Injury. The paper is well written, data are correcly presented and limitation of the study are correctly evidenced.

Author Response

Dear Reviewer 2, Thank you for your review of our manuscript. We have not uploaded a formal cover letter as there were no specific points from you for us to consider. But we appreciate the time you took to review our manuscript.

Reviewer 3 Report

Sigdel et al., performed a retrospective study in 178 unique renal transplant patients using a novel single nucleotide polymorphism (SNP)-based multiplexed PCR (mm PCR) methodology to measure donor-derived cell-free DNA (dd-cf DNA) for the detection of allograft rejection/injury without prior knowledge of donor genotypes. Authors found that median dd-cf DNA was significantly higher in samples with biopsy-proven AR vs. non-AR renal biopsies and conclude that SNP based dd-cf DNA assay detected allograft rejection with superior performance compared with the current standard of care.

The authors have a done a fabulous job in the design of the study with robust methodology, well-described results, and excellent discussion.

There are well over 17 published retrospective and prospective studies on the renal transplant recipients studying the use of dd-cf DNA and monitoring of renal transplant health.

Following are my comments and suggestions to the authors,

Introduction:

In Page 2 of 21, line 61. Authors wrote ‘current treatment options.’ Do they mean current diagnosis options?

-Authors should consider providing some background on the Steady-state, half-life and other details of the dd-cfDNA levels, (1).

Materials and methods:

On page 2 of 21, line 80. Authors wrote ‘Blood was collected from male or female adult or young adult recipients of kidney transplants at various time points following transplantation surgery.’ Can they elaborate on the time points? As some published studies report, the steady-state levels of the dd-cf DNA vary post-transplant and falls rapidly to a baseline level within two weeks.

Results:

On page 10 of 21, line 242. Though only ten biopsies were classified as either TCMR or TCMR and bABMR, did authors notice any differences in the relationship of dd-cf DNA levels between mild TCMR vs. Severe TCMR?

On page 11 of 21, line 268 to 275. Authors wrote ‘Among these STA patients, dd-cf DNA levels were lower at month 0 than subsequent time points. In a systematic review by Knight et al., it was observed that the dd-cf DNA falls rapidly to a baseline level within two weeks of transplantation once the initial ischemia-reperfusion injury has subsided. Can authors describe and discuss why it was different in their study population? Is it because of the ‘novel’ assay they used or something else?

Discussion:

Page 15 of 21. Line 313-315. The authors examined the ability of dd-cf DNA combined with eGFR to predict rejection status (AR/non-rejection) in biopsy matched samples. There is evidence supporting that, the combined use of DSA and dd-cf DNA levels may improve diagnostic accuracy, (3). Did or Can authors perform such analysis in their ABMR patients?

Page 15 of 21, line 326-328. The authors wrote ‘On the other hand, specificity (73%) was slightly lower in the current study, partly driven by the fact that a majority of the “false positives” were cases with BL and OI indicating some form of organ injury. As with available prior literature do authors believe that dd-cf DNA levels could have been influenced by BK virus nephropathy, UTI’s or ATN? Along with any CNI toxicity or interstitial fibrosis or tubular atrophy?

-Can the authors report when were the plasma samples were analyzed (time frame about biopsy)? As dd-cf DNA levels have very short half-lives?

-In the patients who had the renal biopsies, was UTI ruled out?

- There is evidence that elevated dd-cf DNA levels were reported weeks before the clinical diagnosis of acute rejection. As authors collected plasma samples at various time points, any such observation made in the study?

- In some published studies the authors followed dd-cf DNA levels after successful treatment of acute rejection, demonstrating a fall to baseline levels in most cases, (4). Have authors made any such observation as it’s a retrospective study?

- Moriera et al. demonstrated that the simple addition of procalcitonin as a marker of infection improves the specificity of dd-cf DNA in the setting of renal transplant rejection, (5)

Can authors make any comment of their study assay dd-cf DNA levels has any relation to infections?

Thank you for submitting the manuscript to JCM.

References:

1. Beck J, Bierau S, Balzer S, et al. Digital droplet PCR for rapid quantification of donor DNA in the circulation of transplant recipients as a potential universal biomarker of graft injury. Clin Chem. 2013;59:1732-41

2. Lee H, Park Y-M, We Y-M, et al. Evaluation of Digital PCR as a Technique for Monitoring Acute Rejection in Kidney Transplantation. Genomics Inform. 2017;15:2-10.

3. Jordan S, Bunnapradist S, Bromberg J et al. Donor-derived cell-free DNA identifies antibody mediated rejection with graft injury in DSA-positive kidney transplant recipients. Am J Transplant. 2018;18(Supplement 4):254.

4. Mieczkowski P, Malc E, Steele P, Kozlowski T. Human blood cell-free circulating DNA (cfDNA) and miRNA as biomarkers of liver and kidney antibody mediated rejection (AMR) or cellular allograft rejection (ACR). Pilot study. Transplantation 2016; 100(7 Supplement 1):S483.

5. Moreira VG, Garcia BP, Martin JMB, Suarez FO, Alvarez FV. Cell-free DNA as a noninvasive acute rejection marker in renal transplantation. Clin Chem. 2009;55:1958-66.

Author Response

Dear Reviewer 3, Thank you for your review and please see the point-by-point response in the uploaded PDF.

Point 1: In Page 2 of 21, line 61. Authors wrote ‘current treatment options.’ Do they mean current diagnosis options?
Response 1: Yes, thank you. “treatment” has been replaced with “diagnostic” on line 62.
Point 2. Authors should consider providing some background on the Steady-state, half-life and other details of the dd-cfDNA levels, (1).
Response 2: The half-life of cfDNA is much shorter than the time-scale of the inflammation leading to dd-cfDNA release, and thus does not need to be considered when determining the injury status of the renal allograft. Renal allograft levels are reported to be between 0.3% and 1.2% (Knight, et al.). We have added the following underlined clause: “Recently, it was demonstrated that plasma dd-cfDNA fraction, typically between 0.3% and 1.2%, can discriminate active rejection status from stable organ function in kidney transplant recipients” in the introduction at line 63.
citation: Knight SR, Thorne A, Faro MLL, : Donor-specific Cell-Free DNA as a Biomarker in Solid Organ Transplantation. A Systematic Review. Transplantation 2018 Oct 11. doi: 10.1097/TP.0000000000002482. [Epub ahead of print]
MATERIALS AND METHODS:
Point 3. On page 2 of 21, line 80. Authors wrote ‘Blood was collected from male or female adult or young adult recipients of kidney transplants at various time points following transplantation surgery.’ Can they elaborate on the time points? As some published studies report, the steady-state levels of the dd-cf DNA vary post-transplant and falls rapidly to a baseline level within two weeks.
Response 3: Time points were not regular. Additional detail is included in the same paragraph. For example, it states that “time points were typically biopsy-matched at time of clinical dysfunction and biopsy or at the time of protocol biopsy and no clinical dysfunction. In addition, some patients had serial post transplantation blood drawn as part of routine IRB approved bio-sampling studies…72.3% were drawn on the day of biopsy.
For further clarity, information in the paragraph was rearranged to read as follows:
“Male and female adult or young adult patients received a kidney from related or unrelated living donors, or unrelated deceased donors. Plasma samples were obtained from an existing biorepository; time points of patient blood draw following transplantation surgery were irregular. Typically, samples were biopsy-matched at time of clinical dysfunction and biopsy or at the time of protocol biopsy and no clinical dysfunction. In addition, some patients had serial post transplantation blood drawn as part of routine IRB approved bio-sampling studies. The selection of study samples was based on (a) adequate plasma was available, (b) if the sample was associated with biopsy information. Among study samples, 72.3% were drawn on
the day of biopsy. Patients without biopsy-matched samples were excluded from the primary analyses.”
RESULTS:
Point 4. On page 10 of 21, line 242. Though only ten biopsies were classified as either TCMR or TCMR and bABMR, did authors notice any differences in the relationship of dd-cf DNA levels between mild TCMR vs. Severe TCMR?
Response 4: Our analysis did not show a significant difference in dd-cfDNA levels between mild and severe TCMR cases, though the number of samples was quite small. See figure below:

Point 5. On page 11 of 21, line 268 to 275. Authors wrote ‘Among these STA patients, dd-cf DNA levels were lower at month 0 than subsequent time points. In a systematic review by Knight et al., it was observed that the dd-cf DNA falls rapidly to a baseline level within two weeks of transplantation once the initial ischemia-reperfusion injury has subsided. Can authors describe and discuss why it was different in their study population? Is it because of the ‘novel’ assay they used or something else?
Response 5: The review by Knight et al. report that dd-cfDNA levels return to baseline by day 7-10. Our data indicate that dd-cfDNA levels at day 30 are above those at day 0. These time periods are not overlapping, so it is not clear that the two statements are incompatible. We speculate that over time, the dd-cfDNA burden can increase due to accrual of chronic graft injury.
DISCUSSION:
Point 6. Page 15 of 21. Line 313-315. The authors examined the ability of dd-cf DNA combined with eGFR to predict rejection status (AR/non-rejection) in biopsy matched samples.
There is evidence supporting that, the combined use of DSA and dd-cf DNA levels may improve diagnostic accuracy, (3). Did or Can authors perform such analysis in their ABMR patients?
Response 6: Looking at ABMR rejections and DSA we get this plot:

The data in the figure above show that the median dd-cfDNA level is the same for AR patients that are DSA positive vs. negative. Therefore, it does not appear that DSA are correlated with dd-cfDNA.
Point 7. Page 15 of 21, line 326-328. The authors wrote ‘On the other hand, specificity (73%) was slightly lower in the current study, partly driven by the fact that a majority of the “false positives” were cases with BL and OI indicating some form of organ injury. As with available prior literature do authors believe that dd-cf DNA levels could have been influenced by BK virus nephropathy, UTI’s or ATN? Along with any CNI toxicity or interstitial fibrosis or tubular atrophy?
Response 7: Yes, dd-cfDNA levels could have been elevated due to these injuries, but we need more data to confirm this.
Point 8. Can the authors report when were the plasma samples were analyzed (time frame about biopsy)? As dd-cf DNA levels have very short half-lives?
Response 8: Blood was drawn at the same day as biopsy in 217 cases. While cfDNA has short half-life, inflammation leading to cfDNA release is slow to change, so cfDNA levels are not expected to change quickly.
Point 9. In the patients who had the renal biopsies, was UTI ruled out?
Response 9: Yes, UTI was always ruled out before a biopsy was performed. To address this, at line 85-86, we have clarified the test to read: “Typically, samples were biopsy-matched at time of clinical dysfunction and biopsy or at the time of protocol biopsy, at which time most patients did not have clinical dysfunction.” and we have added, at line 94: “Biopsies are not done in cases of active UTI or other infections.”
Point 10. There is evidence that elevated dd-cf DNA levels were reported weeks before the clinical diagnosis of acute rejection. As authors collected plasma samples at various time points, any such observation made in the study?
Response 10: Figure 7C shows the patients with longitudinal samples prior to a biopsy indicating AR. Many of these patients (9/11) have an elevated read of dd-cfDNA prior to the AR biopsy. This is mentioned in line 306-307.
Point 11. In some published studies the authors followed dd-cf DNA levels after successful treatment of acute rejection, demonstrating a fall to baseline levels in most cases, (4). Have authors made any such observation as it’s a retrospective study?
Response 11: Unfortunately, this cohort did not contain any samples taken shortly after treatment of acute rejection.
Point 12. Moriera et al. demonstrated that the simple addition of procalcitonin as a marker of infection improves the specificity of dd-cf DNA in the setting of renal transplant rejection, (5)
Response 12: Unfortunately, proclacitonin levels were not collected on patients in this cohort.
We have added the following sentence at line 342-343: “Combining dd-cfDNA with other markers may provide improved predictive value, but this was outside the scope of this study.”
Point 13. Can authors make any comment of their study assay dd-cf DNA levels has any relation to infections?
Response 13: We are not able to comment, as none of the patients had active infections at the time of biopsy. It is possible that dd-cfDNA levels were elevated due renal injury stemming from infections such as BK nephritis or pyelonephritis, but we do not have data to test this.

Round 2

Reviewer 3 Report

I have reviewed the original manuscript and authors have answered to all my comments and made changes to the paper where suggested.

Thank you for submitting the article to JCM and wish the authors all luck.